# Orbitronics: light-induced orbital currents in Ni studied by terahertz emission experiments

Yong Xu[1,2,3,5], Fan Zhang[3,5], Albert Fert[2,4,5] ✉, Henri-Yves Jaffres [4,5], Yongshan Liu[2,3], Renyou Xu[2,3], Yuhao Jiang[2], Houyi Cheng[2,3] & Weisheng Zhao [1,2,3] ✉

Orbitronics is based on the use of orbital currents as information carriers. Orbital currents can be generated from the conversion of charge or spin currents, and inversely, they could be converted back to charge or spin currents. Here we demonstrate that orbital currents can also be generated by femtosecond light pulses on Ni. In multilayers associating Ni with oxides and nonmagnetic metals such as Cu, we detect the orbital currents by their conversion into charge currents and the resulting terahertz emission. We show that the orbital currents extraordinarily predominate the light-induced spin currents in Ni-based systems, whereas only spin currents can be detected with CoFeB-based systems. In addition, the analysis of the time delays of the terahertz pulses leads to relevant information on the velocity and propagation length of orbital carriers. Our finding of light-induced orbital currents and our observation of their conversion into charge currents opens new avenues in orbitronics, including the development of orbitronic terahertz devices.

The conversion from a charge current $\mathbf{j_C}$ into a spin current $\mathbf{j_S}$ has been studied extensively in systems with strong spin-orbit coupling[1,2]. The most successful example of the conversion mechanism is the spin Hall effect for nonmagnetic metals containing heavy atoms[3–6]. For two-dimensional electron gases at Rashba interfaces and the surfaces or interfaces of topological insulators, the charge-to-spin conversion is usually named the spin Rashba-Edelstein effect[7–10]. In ferromagnet (FM)/nonmagnet (NM) heterostructures, the spin currents induced by spin Hall effect or spin Rashba-Edelstein effect are strong enough to reverse the magnetization of the FM material by spin-orbit torque[11–20]. Therefore, both spin Hall effect and spin Rashba-Edelstein effect have attracted much attention due to their technological significance in developing future magnetic memory devices. Their inverse effects, namely, inverse spin Hall effect[21] and inverse spin Rashba-Edelstein effect[22], convert a spin current $\mathbf{j_S}$ into a charge current $\mathbf{j_C}$. These effects have been widely utilized for detecting spin currents generated by other stimuli, such as the heat current[23], the charge current[5,6], and the spin-pumping technique[24–26]. In recent years, inverse spin Hall effect and inverse spin Rashba-Edelstein effect have also been utilized to generate ultrafast charge pulses and develop efficient broadband spintronic terahertz emitters[27–30].

Several recent works have highlighted the importance of the orbital degree of freedom in condensed matter physics and kicked off the emergent research field of orbitronics[31]. Orbitronics exploits the transport of orbital angular momentum through materials by orbital currents, which can be used as an information carrier in solid-state devices. As for the conversion between spin and charge current by spin Hall effect or spin Rashba-Edelstein effect, it has been theoretically predicted and experimentally shown that a charge current $\mathbf{j_C}$ can be converted into an orbital current $\mathbf{j_L}$ via the orbital Hall effect (OHE) or the orbital Rashba-Edelstein effect (OREE)[32–40].

[1]National Key Lab of Spintronics, International Innovation Institute, Beihang University, Hangzhou 311115, China. [2]Fert Beijing Institute, School of Integrated Circuit Science and Engineering, Beihang University, Beijing 100191, China. [3]Hefei Innovation Research Institute, Beihang University, Hefei 230013, China. [4]Laboratoire Albert Fert, CNRS, Thales, Université Paris-Saclay, Palaiseau 91767, France. [5]These authors contributed equally: Yong Xu, Fan Zhang, Albert Fert, Henri-Yves Jaffres. ✉e-mail: albert.fert@cnrs.fr; weisheng.zhao@buaa.edu.cn

Unlike spin, the orbital current cannot exert a torque directly on the magnetization due to the lack of direct coupling between the orbital current and the magnetization. However, the spin-orbit coupling can convert an orbital current $\mathbf{j_L}$ into a spin current $\mathbf{j_S}$, generating torques on the magnetization. Several studies have identified the spin torques originating from OHE or OREE in heterostructures[37,38].

According to the Onsager reciprocal relation, the inverse effect of the OHE or OREE should convert an orbital current into a charge current. Reliable orbital current sources are still lacking and often covered due to the disturbance of the omnipresent inverse spin Hall effect. However, previous studies show that Ni is a good orbital source compared to other ferromagnets[37,38]. In the same vein, recently, Hayashi et al. demonstrated the generation of orbital current from Ni using the spin pumping technique[41]. In this paper, we provide strong evidence for conversion from a light-induced orbital current $\mathbf{j_L}$ in Ni into a charge current through free-space terahertz emission. Recently, Wang et al. [42] as well as Seifert et al. [43] adopted a similar technique and demonstrated the terahertz emission due to orbital currents. The new effects, the orbital counterpart of inverse spin Hall effect and inverse spin Rashba-Edelstein effect, is named the inverse orbital Hall effect (IOHE) and the inverse orbital Rashba-Edelstein Effect (IOREE)[31]. Thanks to the induced charge current, the IOHE and IOREE provide a feasible approach for the electrical characterization of the orbital current.

Here, we present experiments of THz emission in the time domain generated by the conversion of light-induced spin and orbital currents into ultrafast charges currents in MgO/NM/FM multilayers in which NM = Pt, Ta or Cu and FM = CoFeB or Ni (fabrication and experimental details in Methods). We show that, with Ni, the main contribution to the THz emission originates from orbital currents and can be analyzed to derive information on the velocity and propagation of orbital carriers.

## Results

### Identification of orbital-to-charge conversion

We begin by comparing THz waveforms from MgO (3 nm)/NM (4 nm)/$Co_{0.2}Fe_{0.6}B_{0.2}$ (10 nm)/MgO/Ta and MgO (3 nm)/NM (4 nm)/Ni (10 nm)/MgO/Ta, where NM = Ta, Pt, or Cu (Fig. 1). The NM/CoFeB samples follow the typical behavior of spintronic terahertz emission driven by conversion from spin currents into charge currents via inverse spin Hall effect in NM. The ultrafast charge current induced by the spin current is given as $\mathbf{j_C} = \gamma_S \mathbf{j_S}$, where $\mathbf{j_S}$ refers to the spin current, and $\gamma_S$ is the spin-to-charge efficiency of the NM layer. For a given sample, the waveform polarity shows the expected reversal when the sample is flipped from front excitation to back excitation. Moreover, the waveform polarities of NM/CoFeB, dictated by the sign of the spin Hall effect of NM, are opposite for Ta relative to Pt, as naturally expected from their opposite spin Hall effect[4,5]. In agreement with the very small spin-orbit interaction and negligible inverse spin Hall effect in Cu, the sample CoFeB/Cu shows a small signal very similar to the signal obtained with a CoFeB monolayer without any evidence of additional contribution from Cu. The intrinsic signals in CoFeB monolayers can be ascribed to the anomalous Hall effect (AHE) of the CoFeB layer[44-46], as it will be discussed later.

In contrast, the waveforms of NM (4 nm)/Ni(10 nm) show reversed polarities with respect to Ni monolayer and the same polarities for all

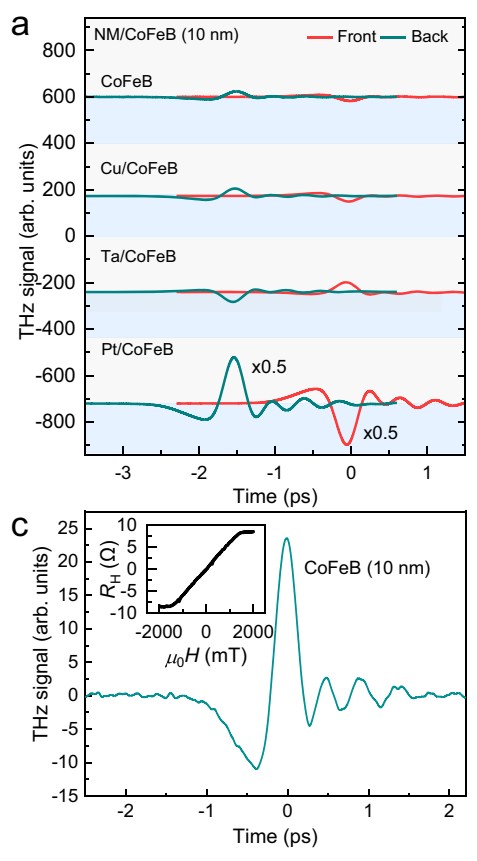

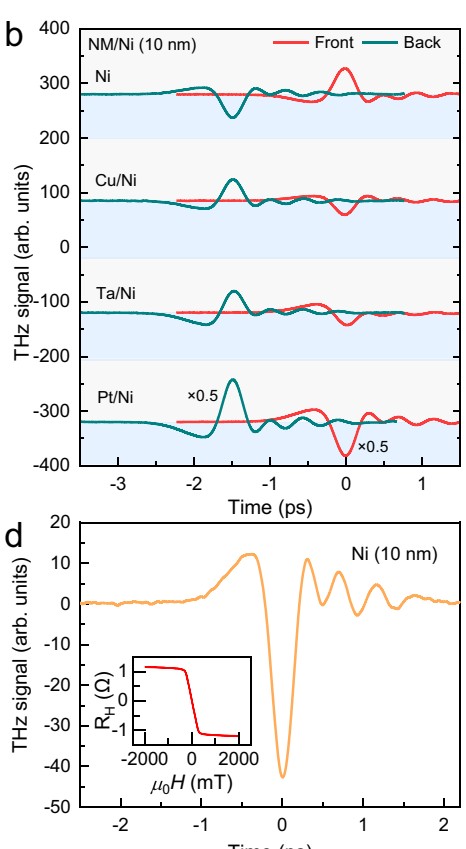

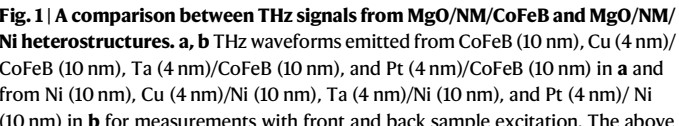

**Fig. 1 | A comparison between THz signals from MgO/NM/CoFeB and MgO/NM/Ni heterostructures. a, b** THz waveforms emitted from CoFeB (10 nm), Cu (4 nm)/CoFeB (10 nm), Ta (4 nm)/CoFeB (10 nm), and Pt (4 nm)/CoFeB (10 nm) in **a** and from Ni (10 nm), Cu (4 nm)/Ni (10 nm), Ta (4 nm)/Ni (10 nm), and Pt (4 nm)/Ni (10 nm) in **b** for measurements with front and back sample excitation. The above experiments were carried out under the same experimental conditions under an in-plane magnetic field of 80 mT. The polarization of the pump laser does not influence the THz emission from Cu/Ni samples (Fig. S3 in Supplementary Information). **c–d** Opposite waveforms for single layers of MgO/CoFeB (**c**) and MgO/Ni (**d**), with insets showing measurements of the opposite AHE in CoFeB and Ni.

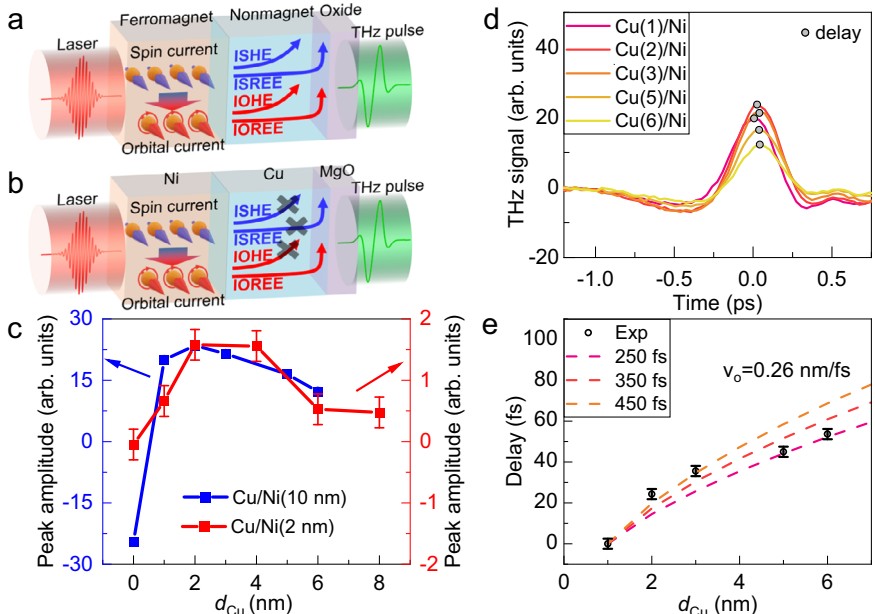

**Fig. 2 | Conceptional diagrams of the spin/orbital-charge conversion and the impact of Cu thickness on THz emission from MgO/Cu/Ni. a** Conceptional diagram of the THz emission showing the different channels for conversion from spin and orbital currents emitted by the ferromagnet into charge currents in the nonmagnetic layer and at the interface with MgO. The AHE channel is not represented. **b** Same diagrams as in **a** for the simpler situation of MgO/Cu/Ni bilayer. The crosses indicate the conversion channels that are negligibly involved in the THz emission. See text on the predominance of the IOREE channel at the Cu/MgO interface. **c** THz peak amplitudes for MgO/Cu/Ni (10 nm) and MgO/Cu/Ni (2 nm) samples as a function of the Cu thickness $d_{Cu}$. The errorbar corresponds to the noise level of the setup. **d** THz emission signal for MgO/Cu/Ni (10 nm) samples for different Cu thicknesses. The maxima of the signals are highlighted by a circular marker. **e** Delay $\tau_D$ versus Cu thickness $d_{Cu}$ from experimental results in Fig. 2d. Curves are calculated from Eq. 2 with the parameters $v_o$ = 0.26 nm/fs and three values of $\tau_{of}$, see inset. The errorbar (vertical line at the datapoints) is estimated based on the multiple scans of a given sample.

three samples, regardless of the sign of spin Hall angle of the NM layers (Fig. 1b). Due to the opposite spin Hall angle of Pt and Ta, spin-charge conversion via inverse spin Hall effect should give opposite polarities for Pt/Ni and Ta/Ni. In addition, as Cu(4 nm)/Ni(10 nm) shows a waveform opposite to the intrinsic one of the Ni monolayer, this reversal reveals a significant additional contribution from Cu in spite of its minimal spin Hall angle. Our results with Ni, the same polarity with Pt and Ta and significant contribution from the addition of Cu, show that inverse spin Hall effect is not sufficient to explain the THz waveforms for Ni-based bilayers.

Since the results of NM/Ni cannot be fully explained with the spin-to-charge conversion by inverse spin Hall effect, we examine the other possible mechanisms. Excluding magnetic dipole emission (first reported by Beaurepaire *et al.* [47]), with generally of smaller amplitude, which cannot give opposite polarities for front and back excitations, we consider THz emission driven by AHE, and THz emission driven by spin-to-charge conversion or orbital-to-charge conversion.

Considering the AHE contribution, the intrinsic emissions by Ni and CoFeB single layers on MgO are predominantly related to AHE with opposite polarities in agreement with the opposite AHE of Ni and CoFeB shown in the inset of Fig. 1c-d. In addition, it is important to note in Fig. 1a that only adding metals with significant spin Hall effect (MgO/Ta/CoFeB, MgO/Pt/CoFeB) changes the signal of the CoFeB monolayer, while adding 4 nm of Cu changes it in a negligible way, as we also checked with other thicknesses of Cu. It means that the contribution from AHE is only slightly affected by adding a nonmagnetic metallic layer (such as Cu) between MgO and a ferromagnetic metal (such as CoFeB, Supplementary Information S5). In MgO/Cu/Ni, as we will discuss, the insertion of Cu introduces a predominant orbital contribution reversing the sign of the emission by AHE in Ni single layers and the dependence of the AHE contribution on Cu thickness is more difficult to isolate. However, we will also use samples with only 2 nm of Ni for which the AHE contribution, as expected and observed for very thin

magnetic layers [46], drops to an extremely small value and can be neglected.

The main result in Fig. 1a is that spin-to-charge conversion explains THz emission in NM/CoFeB bilayers. However, the THz emissions of the MgO/NM/Ni structures are in contradiction with this mechanism in the following aspects: (i)The addition of Ta, Pt, or Cu leads to significant emissions of the same polarity for all three nonmagnetic metals, and this polarity is reversed with respect to the polarity of the AHE-induced emission by Ni monolayers without NM. (ii)Since the spin Hall angle is very small for Cu, this reversal with respect to the AHE-induced signal in the Ni monolayer cannot be explained by the inverse spin Hall effect. (iii)The same polarity of the signals for Ta and Pt is also in contradiction with the conversion of spin current to charge current since Ta and Pt have opposite spin Hall effects.

As in very recent THz results with Ni [43], our results can be explained by a significant light-induced generation of orbital currents in Ni, as illustrated by Fig. 2a. Among the contributions to THz emission, the charge currents generated by orbital-to-charge conversion of these orbital currents in the NM layer dominates over both the AHE intrinsic signal of Ni and the spin-to-charge conversion of the light-induced spin currents. Quantitatively we can write for the resulting charge current:

$$\mathbf{j_C} = \gamma_{AHE}\mathbf{j_I} + \gamma_L\mathbf{j_L} + \gamma_S\mathbf{j_S} \qquad (1)$$

where $\gamma_L$ and $\gamma_S$ are orbital-to-charge and spin-to-charge conversion coefficients. The conversion of the orbital and spin current emitted from the ferromagnetic layer can occur inside the nonmagnetic layer (inverse spin Hall effect or IOHE) or at the interface of the nonmagnetic layer with MgO (inverse spin Rashba-Edelstein effect or IOREE), as illustrated by the 4 possible channels in Fig. 2a. Our results for NM/Ni, with the polarities for Cu, Ta, and Pt aligned in the same direction

(opposite to the direction for a Ni single layer) show the predominance of the orbital terms (bulk IOHE and interfacial IOREE) on spin terms (inverse spin Hall effect and inverse spin Rashba-Edelstein effect) and AHE terms, finally aligning all the polarities. As we have seen, the results with CoFeB present the different signs expected for inverse spin Hall effect in Pt and Ta, and also the absence of a significant signal added by Cu in Cu/CoFeB, in agreement with the very small inverse spin Hall effect expected for Cu. They indicate that CoFeB is much less efficient than Ni for the production of light-induced orbital currents.

### Analysis of THz emission induced by orbital currents emitted by Ni in MgO/Cu/Ni samples

For a quantitative analysis of the different contributions in Eq. 1, MgO/Cu/Ni is our simplest system for the following reasons.

a. Due to the very small spin-orbit coupling of Cu, we can assume that the inverse spin Hall effect in Cu and inverse spin Rashba-Edelstein effect at the MgO/Cu interface have negligible contributions, as expected for inverse spin Hall effect from the negligible calculated Spin Hall Conductivity in Cu[48] and, for inverse spin Rashba-Edelstein effect, because its contribution should have appeared after inserting Cu between MgO and CoFeB (Fig. S5).

b. From ab-initio calculations[48] the OHE of Cu is one of the lowest OHE in metals, whereas large OREE have been observed for the Cu/MgO interface[49] in Co/Cu/MgO trilayers which show the predominant effects of conversion from charge current to orbital current.

c. In addition, the predominant contribution from IOREE at the MgO/Cu interface over IOHE is consistent with the dependence of the emission amplitude on Cu thickness in MgO/Cu/Ni samples (Fig. 2c). For IOREE, one expects the abrupt increase as soon as MgO is covered by Cu, in agreements with Fig. 2c. In contrast, for bulk IOHE, one would expect a progressive increase following the increase of the active region. To avoid the interplay between the contributions from conversion of orbit current (2/3) and AHE (1/3 and opposite signs) in MgO/Cu/Ni (10 nm), we studied a similar series with Ni (2 nm) for which we found a negligibly small AHE-induced THz emission from a single layer of Ni 2 nm, as expected and observed in ultra-thin metallic films[46] (Section 4 of Supplementary Information). For MgO/Cu/Ni (2 nm) in Fig. 2c, the amplitude is about zero for $d_{Cu} = 0$ (negligible AHE contribution with Ni 2 nm), and jumps to about its maximum value as soon as a thin layer of Cu (2 nm) is deposited to cover almost totally MgO and then remains at about the same level. The progressive decay above 4 nm is due to the change in laser absorption and thin film impedance. For MgO/Cu/Ni (10 nm) on the same figure, the negative amplitude at $d_{Cu} = 0$ expresses the negative contribution from AHE and the positive contribution introduced by Cu reverses the signal as soon as the thinnest Cu layer (1 nm) is deposited to fully cover MgO before a progressive decrease above 2 nm. The difference with respect to the results for Ni (2 nm) is that this decrease can be due to the variation of the film impedance (light absorption) as well as the variation of the AHE contribution (although the results presented in Supplementary Information S5 for MgO/Cu/CoFeB seem to indicate that the insertion of Cu does not change significantly the AHE, at least with CoFeB).

So, the main result is that the signal abruptly reached at its maximum value by covering MgO with Cu, with Ni 2 nm as well as with Ni 10 nm, is not consistent with a progressive introduction by IOHE in the bulk of Cu and in favor of interfacial IOREE, as expected by theory and observed by Kim et al.[49] in several types of trilayers including Co/Cu.

Recent experiments of THz emission by Ni/W bilayers have also revealed the light-induced emission of orbital currents by Ni[43] and, for the emission, the predominance of orbit to charge at the interface between W and SiO₂. In Ni/W/SiO₂ stacks, their interfacial conversion

by IOREE at the W/SiO₂ interfaces introduces time delays $\tau_D$ time shifts in the waveforms of THz emission related to the orbital carrier flying time through W. An interesting information can be then derived from the variation of these delays as a function of W thickness, with a crossover from a linear initial variation in a ballistic regime to a quadratic initial variation in the diffusive regime (Supplementary Information of[35]). As the small signals obtained with MgO/Cu/Ni (2 nm) are too small for a precise analysis, we consider only the delays for Ni (10 nm). The results in Fig. 2d (frequency spectra shown in Supplementary Information S6) rule out an interpretation in the diffusive regime which would lead to a quadratic increase with thickness[25,50] and the linear initial variation indicate that the transport in Cu is ballistic[43]. In the ballistic regime, we extended the formalism in[43] by introducing a probability of orbital flip (or orbital decoherence) that we characterize by the orbital flip time $\tau_{of}$, (see Methods and Supplementary Information S1). The probability of orbital flip, even small, introduces deviations from the linear variation of the delay as a function of the thickness of Cu, see the concavity of the curves in Fig. 2e (a reference system without time shift is shown in Supplementary Information S7). The delay is expressed as a function of $\tau_{of}$ and $t_0 = \frac{d_{Cu}}{v_o}$, where $v_o$ is the group velocity of the orbital carriers, in the following expression:

$$\tau_D = \frac{\int_{t_o}^{\infty} dt \, \exp\left(-\frac{t}{\tau_{of}}\right)}{\int_{t_o}^{\infty} \frac{dt}{t} \, \exp\left(-\frac{t}{\tau_{of}}\right)} \tag{2}$$

As shown in Fig. 2e, the curves calculated from Eq. 2 show a qualitative agreement with the experimental variation for the deviation from linearity and for the concavity (We made the approximation that the AHE contribution in Ni 10 nm could not affect significantly the dependence of the delay on the Cu thickness). The typical orbital group velocity of carriers extracted from our fit (see Supplementary Information S2), $v_o \approx 0.26 \pm 0.08$ nm/fs, appears to be an order of magnitude below the charge carrier velocity in Cu at room temperature. This striking result is not really surprising because charge currents and orbital currents cannot be carried by the same type of carriers and same type of bands in the crystal. In the particular case of Cu, one knows that the charge current is predominantly carried by s-band electrons (no orbital moment) whereas orbital currents are necessarily carried by carriers of s-p hybridization with enhanced effective mass and possibly carriers at higher energy induced by light.

For the orbital flip time $\tau_{of}$, we obtain an idea of its order of magnitude corresponding to the qualitative agreement obtained for $\tau_{of}$ between 250 and 450 fs. As shown in Supplementary Information S1, from the information on $\tau_{of}$ obtained in the ballistic transport regime of THz experiments, we can derive the expected orbital diffusion length in the usual diffusive picture of dc orbital currents: $l_{of} = v_o \sqrt{\frac{\tau_{of} t_p}{3}}$, where $t_p$ is the typical scattering momentum relaxation time (typically 10 fs for Cu at room temperature). Therefore, the orbital diffusion length in Cu may be estimated to be 9 nm. This short orbital diffusion length is consistent with the experiments of orbital transport through Cu by Kim et al.[49] in Co/Cu/Oxide with a drop of the efficiency for Cu thicker than about 15 nm. This is much shorter than what has been found for the orbital diffusion length in d-band metals, for example, around 50 nm or 70 nm in Ti[40,51]. As for the orbital carrier velocity, the short orbital diffusion length in Cu expresses the s character of the conduction band and the orbital transport only by states with s-p hybridization.

To sum up, the analysis of the results in the simplest situation of MgO/Cu/Ni leads to two main results: 1) THz emission due predominantly to the conversion of orbital current by IOREE at MgO/Cu interface. 2) The analysis of the delays of the waveforms of carefully driven THz emission experiments provides interesting data on the fundamental dynamics of the light-induced orbital carriers emitted by Ni, namely the orbital carrier velocity $v_o$ and the orbital flip time $\tau_{of}$. In

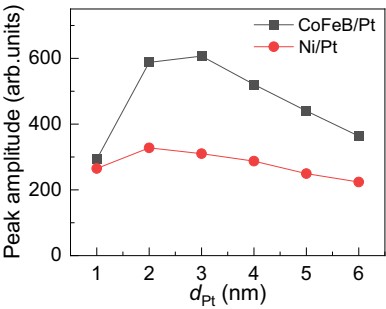

**Fig. 3** | THz peak amplitudes from raw data of Ni (5 nm)/Pt and CoFeB (5 nm)/Pt as a function of the Pt thickness measured with front excitation.

light of the complexity arising from different contributions, we remind the reader that our theoretical discussion considered only the main contribution, the orbital-to-charge conversion at the Cu/MgO interface.

### THz emission combining contributions from spin and orbital currents

We now consider THz emission in systems with added contributions from conversions of both spin currents (by inverse spin Hall effect and inverse spin Rashba-Edelstein effect) and orbital currents (by IOHE and IOREE), as in the example of samples with Pt for the NM layer and CoFeB or Ni for the FM layer in Fig. 3. With CoFeB, as shown by the opposite signals in Fig. 1a for opposite signs of inverse spin Hall effect in Pt and Ta or W, one knows that a predominant contribution comes from the spin component (inverse spin Hall effect). The THz signal with CoFeB in Fig. 3 increases rapidly with Pt thickness up to a maximum at 3 nm of Pt, which is consistent with the progressive increase of the inverse spin Hall effect active region at increasing thickness. The slow decrease after the maximum is what is expected from the increase of light and THz absorption. With Ni, the dependence on Pt thickness (red curve) is different and, qualitatively, appears as a variation with thickness similar to what is observed with CoFeB (black curve) on top of a plateau, reminding the plateau observed with Cu/Ni in Fig. 2c and ascribed to the interfacial conversion of the orbital current emitted by Ni. Precise separation of the contributions of the different conversion mechanisms, inverse spin Hall effect, inverse spin Rashba-Edelstein effect, IOHE, and IOREE illustrated in Fig. 2b would be possible with an extensive series of new types of samples in which the ratio between spin and orbital current can be quantitatively controlled for example, as in[52] by insertion of the rare-earth ultra-thin layer to induce conversions between spin and orbital current and varying their respective values, what is out of the scope of the present paper.

## Discussion

The usual mechanisms of the generation of orbital currents are conversions from charge currents by OHE or OREE or conversions from spin currents. The most striking result of our work is the light-induced generation of orbital current in Ni, in agreement with results recently obtained by Seifert *et al.* [43]. We could not find any theoretical work predicting the production of orbital current by Ni from high-energy excitation. For low energy electrons, according to the coefficients calculated for the conversion between orbital and spin current in Fig. 2b in[37], we can see that the orbital-to-spin conversion is definitely more efficient in Ni compared to other 3d metals or alloys, which shows some similarity with our results. For high-energy excitation, a first tentative scenario is the production of orbital current from the conversion of light-induced spin current, as described for interconversions between orbital and spin in the Fermi energy range[53]. In another scenario, the light-induced production of orbital current might be directly from the excitation of orbital states[54]. Further

theoretical works are needed to study the respective contributions of these two types of mechanisms.

From an experimental point of view, the specific character of Ni for the production of orbital current is confirmed by other types of experiments[37,40]. In Spin Torque FerroMagnetic Resonance (ST-FMR), Ni is described as "abnormal" from the results presented by Lee et al. [37] in their Fig. 3 on the damping-like torque on ferromagnetic metals (CoFeB, FeB, Ni) deposited on Pt or Ta (opposite spin Hall effect). Opposite torques for bilayers of CoFeB (or FeB) with Pt and Ta are observed, as expected from the opposite spin Hall effect of Pt and Ta, whereas the torque on Ni has the same sign with Pt and Ta. This striking similarity between ST-FMR[37,40], orbital pumping by ferromagnetic resonance (FMR)[41], and light-induced THz emission supports the idea that orbital currents can be generated not only by conversion from charge or spin currents but also by different types of excitations, including excitation by light and FMR. Recently, the existence of orbital current generated by FMR of YIG ($Y_3Fe_5O_{12}$) and converted into voltage by IOREE at Pt/Cu/$O_x$ interfaces was also observed[55].

In summary, orbitronics is a promising field of research in which we have presented results obtained by a new experimental method based on the production of light-induced orbital current and their exploitation for terahertz emission in multilayers associating Ni with nonmagnetic metals NM and MgO. We find terahertz emission of the same polarity with Cu, Ta, and Pt for NM despite the opposite spin Hall effect of Ta with respect to Pt (this common polarity is also opposite to the polarity of the AHE-induced emission by a Ni monolayer). We ascribe these results to an efficient light-induced emission of orbital currents in Ni, and we have presented a detailed analysis of them in the simpler situation of MgO/Cu/Ni from which, in addition, we can extract information on the velocity and orbital flip time of the orbital carriers. The main general results are that orbital currents can be generated not only by conversion from charge or spin currents but also by light and, as we noted, also by microwaves in experiments of FMR type. In THz emission experiments, the analysis of the emission delays can bring precious information on the dynamics of the orbital carriers. All these results open new routes for orbitronics and future orbitronic devices.

## Methods
### Experiment procedures

Magnetic multilayer samples were prepared by high-vacuum magnetron sputtering on a glass substrate. Unless otherwise specified, all the samples are deposited on a 3 nm MgO buffer and protected with a capping layer of a MgO(2 nm)/Ta (2 nm) bilayer. In the THz time-domain spectroscopy, femtosecond pulses of 35-fs pulse duration, 1 kHz repetition rate, and 0.25 mJ/cm² fluence are used to pump the magnetic multilayers. We use a lock-in SR830 to measure the signal which is modulated by a chopper of ~200 Hz. The lock-in SR830 extracts the signal at the modulation frequency, with a detection bandwidth inverse proportional to the time constants. Typically, the time constant is set to 300 ms. An interval of 1000 ms is adopted to stabilize the mechanical movement of the delay line.

### Data analyses

One of the strong interests of THz-TDS in the pulsed regime over other techniques is indeed its ability to probe the time dynamics as well as a certain time delay of the spin and/or orbital injection process close to interfaces, presently Ni/Cu and Cu/MgO after an ultra-short pulse generation[43]. Such delay time depends on the Cu thickness ($d_{Cu}$) and the group velocity of orbital carriers ($v_o$) according to $\tau_D(\cos\theta) = \frac{1}{V_o\cos\theta}d_{Cu}$ in the case of a ballistic transport ($d_{Cu}$ smaller than the mean free path, MFP $\lambda$). Concerning the transport properties, one can make the classification below:

- a ballistic transport in the limit of a short coherence time where the traveling distance is linearly proportional to the delay time $\tau_D \sim d_{Cu}$. This case is described in large details in ref. 35.

- a diffusive transport where the average escape distance from the origin increases as the square root of the time according: $d_{Cu} \sim \sqrt{D\tau_D}$ (linked to the diffusion constant D). This leads to a convex shape of the time delay $\tau_D \sim (d_{Cu})^2$ vs. the Cu thickness $d_{Cu}$.
- a ballistic transport accompanied by a long decoherence process where the time delay increases with the distance to the origin as: $\tau_D \sim (d_{Cu})^\alpha$ with ($\alpha < 1$) associated to a concave shape of the corresponding dependence of $\tau_D \, vs. d_{Cu}$.

Note that, in the case of hot carrier transport, there should be an intermediate regime of transport, the so-called *superdiffusive* regime, intermediate between purely ballistic and diffusive regime ($1 < \alpha < 2$) corresponding to excitation of secondary source of electrons. Electrons may transfer their energy to other electrons occupying states below the Fermi energy.

On the other hand, systematic measurements of the resistivity of sputtered Cu layers in a different type of multilayers and the thickness range of our samples led us to estimate the resistivity MFP to about 7 nm (Supplementary Information of[56]), however that it should be slightly longer if we have not to consider the scattering on interfaces involved in the resistivity MFP. It means that our experiments are in the ballistic regime (at the limit of the transition to diffusive), explaining that we do not observe any square dependence of $\tau_D$ on $d_{Cu}$ in Fig. 2e.

## Data availability
The data generated in this study are provided within the article and in the Supplementary Information.

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

## Acknowledgements

The authors gratefully acknowledge the National Key Research and Development Program of China (No. 2022YFB4400200 (W.Z.), No. 2023YFF0719200 (Y.X.)), National Natural Science Foundation of China (No. 92164206 (W.Z.), 62105011(F.Z.), 52261145694 (W.Z.) and 52121001 (W.Z.)). We acknowledge the ANR program ORION through Grant No. ANR-20-CE30-0022-02 (H.J.). All authors sincerely thanks Hefei Truth Equipment Co., Ltd for the help on film deposition. This work was supported by the New Cornerstone Science Foundation through the XPLORER PRIZE (W.Z.).

## Author contributions

W.Z. initialized, conceived and supervised the project. W.Z. Y.X. F.Z. and R.X. conceived the experiments. Y.L. and H.C. deposited the heterostructures. F.Z. carried out the THz measurements. A.F. Y.X. and H.J. analyzed the experimental data. Y.X. F.Z. H.J. and A.F. wrote the manuscript together. All authors discussed the results and commented on the manuscript.

## Competing interests

The authors declare no competing interests.
