## [Peer Review File · Nature Communications]

Reviewers' Comments:

Reviewer #1:

Remarks to the Author:

The manuscript submitted by Y. Xu et al. studies anomalies in the spintronic THz emission experiment by the presence of orbital currents. For two series of samples, namely NM/CoFeB and NM/Ni (NM = none, Cu, Ta, and Pt) series, the anomalies of the sign and magnitude of the emission are found in the NM/Ni series, which cannot be attributed to neither of the inverse spin Hall effect of the NM nor the anomalous Hall effect in Ni. Meanwhile, the NM/CoFeB series exhibits a typical behavior expected from the inverse spin Hall effect in the NM or the anomalous Hall effect of CoFeB. From these results, the authors conclude that appreciable magnitude of orbital currents are generated from Ni, which is less pronounced in CoFeB. The authors additionally analyze the diffusive process of orbital currents by analyzing the waveform of the THz dynamics in detail and compare it to a model.

I find this work interesting enough and well-written for the wide readership of Nature Communications. Also, the analysis of the THz dynamics is inspiring. The field of orbitronics started to receive lots of attention in recent years with the discovery of orbital currents and its application to spintronics. However, most of the experiments addressed the steady state transport in a dc regime, and studies on THz dynamics are found in only a few references. The THz spectroscopy is a tool that are expected to reveal much of microscopic processes of orbital dynamics and transport in the femto-second timescale. Although there are two similar works published recently [P. Wang et al. npj Quantum Materials 8, 28 (2023); T. S. Seifert et al. Nature Nanotechnology (2023), <https://doi.org/10.1038/s41565-023-01470-8>], I still recommend this work to Nature Communications because this work addresses different aspects of THz dynamics such as orbital-flip scatterings in the diffusive transport compared to the two published works. Moreover, all these works appeared on arXiv around the same time, so I assume that independent experiments were carried out at different places without knowing each other's works.

Although I agree with the authors' claim and analysis to a large extent, I find arguments that are still questionable. I also have a few questions because I do not understand completely the authors' arguments in a few places, which may not be straightforward to non-expert readers. I also have a few comments on the manuscript itself.

1. The title of the paper should be more specific on the work that the authors actually did.

The present title may confuse the reader that this work is an extensive analysis of all types of THz emission by orbital currents or a comprehensive review.

2. On line 56, the authors state "To date, few studies reported the conversion from orbit current to charge current. This is due to the lack of a reliable orbit current source ...". Although there are not "many", there are a few papers which addressed the inverse process. The following papers can be cited here: P. Wang et al. npj Quantum Materials 8, 28 (2023); T. S. Seifert et al. Nature Nanotechnology (2023), <https://doi.org/10.1038/s41565-023-01470-8>; H. Hayashi and K. Ando, arXiv:2304.05266. Also, it has been known for many years that Ni is a good orbital source compared to other ferromagnets, as demonstrated by Refs. 29, 30, 35. Another reference, H. Hayashi et al. Communications Physics 6, 32 (2023), suggest the same. I recommend the authors to explicitly cite these references in the introduction.

3. There is no explanation on Figure 1c,d. I understand that the peak amplitude can be simply taken from the THz wave form, but I think the decomposition into several microscopic contributions is highly nontrivial. I do not think we can assume that there is no contribution by orbital currents in the NM/CoFe series (Fig. 1c). In general, the THz emission should be understood as a result of the competition among the contributions by different mechanisms, including spin and orbital contributions. Also, I do not understand how one can simply separate the spin and orbital contributions in Fig. 1d. This requires quantification of various microscopic processes, first, how much spin and orbital currents are generated by the pump laser, second, how much of them are lost at the interface and relaxed during the propagation, and third, how much spin and orbital currents will be converted into the charge current via the inverse spin and orbital Hall effects or Edelstein effects, respectively. The first and third processes may be assumed to be determined by

the stand-alone FM and NM, respectively, but the second process is generally dependent on interface properties, which needs to be characterized case by case. I think the authors should transparently show how to decompose different contributions in Figs. 1c and 1d. Otherwise, I would recommend removing them from the manuscript.

4. The authors claim that a dominant contribution to the THz emission in Ni/Cu/MgO is the inverse orbital Edelstein effect at Cu/MgO interface. However, I do not understand how the orbital angular momentum can be transported through the pure Cu layer whose energy bands near the Fermi energy have mainly s-character.

5. The section on "Analysis of THz emission..." seems not so easy to follow at many places, at least to me. For example, at line 189, the authors say "The observation of an almost linear variation of the delay τ_D in the inset of Fig. 2d", but the inset is missing in Fig. 2d.

Also, it must be explained enough that how the group velocity of the carriers, $v_F \approx 0.26$ nm/fs is obtained. Is it the result of the fitting the curve (least-square, etc.)? If so, what is the value of the uncertainty of this estimation?

On line 197, it is written that "Another striking result... ". I do not understand what the authors had in mind on the concave shape. This needs to be more specific. Thus, I do not understand how it is not the expected behavior from the diffusive transport, which also needs to be mentioned.

Finally, the authors adopt the method of analysis from Ref. 35, which needs to be mentioned. Also, it needs to be specified that how much the analysis in this manuscript is different from or share some common features with that of Ref. 35.

Overall, I think this section needs to be written in a clearer way.

5. It needs to be clarified whether the polarization of the pump laser influences the result. For example, see T. J. Huisman et al. Nature Nanotechnology 11, 455 (2016).

6. On the terminology, I think using "orbital current" might be a better choice instead of "orbit current" because the community already uses the term "orbital current" in many papers.

Reviewer #2:

Remarks to the Author:

Authors: Yong Xu et al.

Title: Orbitronics: Light-induced Orbit Currents in Terahertz Emission Experiments

In this work the authors study THz emission from spintronic heterostructures. By comparing CoFeB based and Ni based stacks and in combination with various normal metals, the authors identify unexpected signals which they attribute to orbital momentum currents and to orbital-to-charge conversion. This work is extremely timely, the data is clean and nicely presented and the results will be of high interest for the community. This type of work would fit perfectly the scope of Nature Communications. However, there are a number of sections of the manuscript that I believe are not entirely clear and should be addressed before consideration for publication:

1. On line 167 it is said « AHE contribution (IS) negligibly dependent on (THE) Cu thickness as supported by the negligible departure from the signal with CoFeB monolayer in Cu/CoFeB samples with different Cu thicknesses in our thickness range (Supporting information and referred to above). » I have a few comments concerning this sentence:

1a. First of all the wording of the sentence is not very clear. What does "negligible departure from the signal" mean here? I'm assuming they mean the signal is constant in Cu/CoFeB samples with different thicknesses, which they attribute to the AHE-THz emission, and they assume it should be the same for Cu/Ni... is it?

1b. Where is the Cu/CoFeB data? It is not in the Supporting information as indicated in the text. Is it just the Cu/CoFeB vs CoFeB monolayer data from Fig1? Two data points is not a lot to claim an "independence" on thickness...

1c. What is the physical understanding of a Cu thickness independent AHE-driven THz emission? That is pretty surprising a priori, since adding metal to the stack should change drastically the THz (absorption) emission. Do the authors expect the AHE to increase with thickness, and therefore to balance out the change in impedance of the stack? If not, what is the intuitive way of understanding such an independence? Would this be the same for Ni and CoFeB samples? Could it be that in the Cu/CoFeB layers there is also orbital currents going on? The assumptions made in the paper are pretty strong, and not discussed enough. In the present form I believe that the amplitude of the orbital effect cannot be properly separated from that from the AHE, and unless having more convincing evidence the thickness dependence of the orbital effect can only be speculatively discussed.

1d. The conclusion on line 169 is not entirely obvious in my opinion. I believe it is a far stronger argument the one used earlier (lines 121-131...) where the polarities expected from SHE and other effects are discussed, and used to suggest that an additional phenomena is required (such as orbital currents).

2. On line 182 the authors say "with the expected additional slow decrease with thickness due to increasing light absorption". Have the authors estimated this decrease? They claim it is "slow", but how slow is it? Is it compatible with a ~50% loss of signal per 6nm of Cu as their data suggests? From the literature, I would say it is much more...

3. My final point is on the uncertainty in the reported delay times in Fig2.e. The measurements have no X or Y error bars (neither in Fig2d nor Fig2e). The delay time seems to be "picked" by choosing the data point at the highest amplitude and not by any fitting function. Also, the fact that a 35fs pulse is being used to extract data points with a ~5fs time-resolution is not discussed. In fact, if I put an error bar equal to half the FWHM (~17fs) onto the data points in Fig.2e I could fit almost any function. The authors should add error bars if they want to discuss such measurements and should discuss the technical details to achieve such a reduced uncertainty. A few points the authors might want to clarify: The data points in Fig2.d the result of single traces or are they the result of multiple scans, in which case could they show the statistics? Have the authors carried some noise characterization in their setup (amplitude and timing noise)? How many time constants did the authors wait before pulling the measurements from the lock-in, and what filters (or ENBW) were they using? Are the measurements independent of delay-line speed, or in other words did the authors carefully verified that the measurements at each position are independent of previous positions? One neat way of demonstrating that the delay line and lock-in timings are just right is to perform the experiments by randomizing the order of time-delays to measure (for ex: measuring first -1ps, then +0.5ps, 0ps, -0.3ps... and so on randomly, instead of going from -2ps to +0.5ps in incremental order). I believe that all of these points could potentially strongly affect the confidence in the observed delays. With proper error bars, a discussion becomes possible which would, indeed, be very interesting.

**REVIEWER COMMENTS**

Reviewer #1 (Remarks to the Author):

The manuscript submitted by Y. Xu et al. studies anomalies in the spintronic THz emission
experiment by the presence of orbital currents. For two series of samples, namely NM/CoFeB and
NM/Ni (NM = none, Cu, Ta, and Pt) series, the anomalies of the sign and magnitude of the emission
are found in the NM/Ni series, which cannot be attributed to neither of the inverse spin Hall effect
of the NM nor the anomalous Hall effect in Ni. Meanwhile, the NM/CoFeB series exhibits a typical
behavior expected from the inverse spin Hall effect in the NM or the anomalous Hall effect of
CoFeB. From these results, the authors conclude that appreciable magnitude of orbital currents are
generated from Ni, which is less pronounced in CoFeB. The authors additionally analyze the
diffusive process of orbital currents by analyzing the waveform of the THz dynamics in detail and
compare it to a model.

I find this work interesting enough and well-written for the wide readership of Nature
Communications. Also, the analysis of the THz dynamics is inspiring. The field of orbitronics
started to receive lots of attention in recent years with the discovery of orbital currents and its
application to spintronics. However, most of the experiments addressed the steady state transport in
a dc regime, and studies on THz dynamics are found in only a few references. The THz spectroscopy
is a tool that are expected to reveal much of microscopic processes of orbital dynamics and transport
in the femto-second timescale. Although there are two similar works published recently [P. Wang et
al. npj Quantum Materials 8, 28 (2023); T. S. Seifert et al. Nature Nanotechnology
(2023), <https://doi.org/10.1038/s41565-023-01470-8>], I still recommend this work to Nature
Communications because this work addresses different aspects of THz dynamics such as orbital-
flip scatterings in the diffusive transport compared to the two published works. Moreover, all these
works appeared on arXiv around the same time, so I assume that independent experiments were
carried out at different places without knowing each other's works.

Although I agree with the authors' claim and analysis to a large extent, I find arguments that are still
questionable. I also have a few questions because I do not understand completely the authors'
arguments in a few places, which may not be straightforward to non-expert readers. I also have a
few comments on the manuscript itself.

1. The title of the paper should be more specific on the work that the authors actually did.
The present title may confuse the reader that this work is an extensive analysis of all types of THz
emission by orbital currents or a comprehensive review.

**Response:**

We agree the title should be less general and indicate that our work is on orbital currents
induced by light in Ni. Our new title is:
Orbitronics: Light-induced Orbit Currents in Ni Studied by Terahertz Emission Experiments

2. On line 56, the authors state "To date, few studies reported the conversion from orbit current to
charge current. This is due to the lack of a reliable orbit current source ...". Although there are not

“many”, there are a few papers which addressed the inverse process. The following papers can be
cited here: P. Wang et al. npj Quantum Materials 8, 28 (2023); T. S. Seifert et al. Nature
Nanotechnology (2023), <https://doi.org/10.1038/s41565-023-01470-8>; H. Hayashi and K. Ando,
arXiv:2304.05266. Also, it has been known for many years that Ni is a good orbital source compared
to other ferromagnets, as demonstrated by Refs. 29, 30, 35. Another reference, H. Hayashi et al.
Communications Physics 6, 32 (2023), suggest the same. I recommend the authors to explicitly cite
these references in the introduction.

**Response:**

We thank the reviewer for summarizing the references. We shifted or added all these references
in the introduction, see lines 57-60 and 61-63.

Lines 57-60:

*However, previous studies show that Ni is a good orbital source compared to other*
*ferromagnets [29,30]. In the same vein, recently, Hayashi et al. demonstrated the generation of orbital*
*current from Ni using the spin pumping technique [33].*

Lines 61-63:

*Recently, Wang et al. [34] as well as Seifert et al. [35] adopted a similar technique and*
*demonstrated the terahertz emission due to orbital currents.*

3. There is no explanation on Figure 1c,d. I understand that the peak amplitude can be simply taken
from the THz wave form, but I think the decomposition into several microscopic contributions is
highly nontrivial. I do not think we can assume that there is no contribution by orbital currents
removed in the NM/CoFe series (Fig. 1c). In general, the THz emission should be understood
as a result of the competition among the contributions by different mechanisms, including spin and
orbital contributions. Also, I do not understand how one can simply separate the spin and orbital
contributions in Fig. 1d. This requires quantification of various microscopic processes, first, how
much spin and orbital currents are generated by the pump laser, second, how much of them are lost
at the interface and relaxed during the propagation, and third, how much spin and orbital currents
will be converted into the charge current via the inverse spin and orbital Hall effects or Edelstein
effects, respectively. The first and third processes may be assumed to be determined by the stand-
alone FM and NM, respectively, but the second process is generally dependent on interface
properties, which needs to be characterized case by case. I think the authors should transparently
show how to decompose different contributions in Figs. 1c and 1d. Otherwise, I would recommend
removing them from the manuscript.

**Response:**

We agree with the recommendation by the referee of removing the diagrams representing
possible decompositions of the different contributions to the results in Fig. 1a-b. We had forgotten
to specify that it was simply a qualitative illustration to help the reader imagine a possible
decomposition of the results in Fig. 1a-b. It is clear that our paper does not present any quantitative
decomposition of all the results in Fig. 1a-b and is centered on the analysis of the simplest system
MgO/Cu/Ni: identification of a predominant contribution from IOREE at Cu/MgO interface and
analysis of its dynamical aspects (delays) with results on the orbital velocity, orbital flip time and
orbital diffusion length in Cu (analysis inspired by the work of Seifert et al but going a little further,
Eqn.2, by the introduction of a small probability of orbital flip in the ballistic regime). Fig.1c and d

keep only the Anomalous Hall Effect measurements explaining the opposite waveforms from AHE
with single layers of CoFeB and Ni.

4. The authors claim that a dominant contribution to the THz emission in Ni/Cu/MgO is the inverse
orbital Edelstein effect at Cu/MgO interface. However, I do not understand how the orbital angular
momentum can be transported through the pure Cu layer whose energy bands near the Fermi energy
have mainly s-character.

**Response:**

The reviewer raised a very important fundamental point: propagation of orbital current through
a metal in which the band near Fermi energy is mainly s with some hybridization with other bands.
However, for Cu the s-p hybridization is quite strong in the conduction band and particularly at the
vicinity of the Fermi level (see the band structure below from Papaconstantopoulos, D. A. Handbook
of the Band Structure of Elemental Solids: From Z = 1 To Z = 112. (Springer US, 2015).)

Figure R1 Energy bands and density of states of Cu. From Papaconstantopoulos, D. A. Handbook of the Band Structure of Elemental Solids: From Z = 1 To Z = 112. (Springer US, 2015).

Experimentally, several experiments show that orbital current can be carried through Cu, for example in the systematic studies of orbital current through Cu between a ferromagnetic metal and oxides (MgO, SiO₂, TiO₂, Al₂O₃) with the Cu/oxide interface acting as orbital sources, orbital transport through Cu detected by the orbital torque on the ferromagnetic metal and characterized by the efficiency coefficient θ [Kim et al. arXiv 2307.09824 (2023), Kim et al., PRB 103, L020407 (2021), group of Otani]. Their results (see Fig. R2) show that the propagation through Cu does not go above about 20 nm, which correspond to an orbital diffusion length of the order 10-15 nm. This length is much-much shorter than it has been found with metal with p or d states around Fermi level, for example 68 nm in α -W or 47 nm in Ti [Hayashi, H. et al. Commun Phys 6, 1 (2023)]. Those results show that a metal of predominantly s character (such as Cu) can transmit an orbital current but not very efficiently. In fact, the results in the figure below are consistent with the Cu orbital diffusion length around 9 nm derived in our paper. Moreover, theory and calculations show the possibility of promoting orbital transport through Cu interfaced with an oxide by Go et al. (D. Go et al., Phys. Rev. B 103, L121113 (2021)).

Figure R 2 Orbital efficiency of several Cu/oxide interface in Ni/Cu/Oxide trilayers (the orbital current is created by conversion of charge current at Cu//oxide interface and detected by torque on Ni). From Kim, arXiv.2307.09824 (2023).

We added a few words on this point in text of the new version.

Line 228-231

In the particular case of Cu, one knows that the charge current is predominantly carried by s-band electrons (no orbital moment) whereas orbital currents are necessarily carried by carriers of s-p hybridization with enhanced effective mass and possibly carriers at higher energy induced by light.

Line 239-244:

This short orbital diffusion length is consistent with the experiments of orbital transport through Cu by Kim et al. [41] in Co/Cu/Oxide with a drop of the efficiency for Cu thicker than about 15 nm. This is much shorter than what has been found for the orbital diffusion length in d-band metals, for example, around 50 nm or 70 nm in Ti [32,42]. As for the orbital carrier velocity, the short orbital diffusion length in Cu expresses the s character of the conduction band and the orbital transport only by states with s-p hybridization.

5. The section on “Analysis of THz emission...” seems not so easy to follow at many places, at least to me. For example, at line 189, the authors say “The observation of an almost linear variation of the delay τ_D in the inset of Fig. 2d”, but the inset is missing in Fig. 2d.

Also, it must be explained enough that how the group velocity of the carriers, $v_F \approx 0.26$ nm/fs is obtained. Is it the result of the fitting the curve (least-square, etc.)? If so, what is the value of the uncertainty of this estimation?

Response:

The values of the group velocity v_o and the orbital flip time τ_{of} are derived from the general formula of the time delay derived from Eq.2:

$$\tau_D = \frac{\int_{t_o}^{\infty} dt \exp\left(-\frac{t}{\tau_{of}}\right)}{\int_{t_o}^{\infty} \frac{dt}{t} \exp\left(-\frac{t}{\tau_{of}}\right)}$$

From the fitting of the experimental data in Fig. 2e, we obtained $v_o = 0.26 \pm 0.08$ nm/fs and, for the orbital flip time τ_{of} related to the concavity of the curves, we obtain an idea of its order of

153 magnitude corresponding to the qualitative agreement obtained for τ_{of} between 250 and 450 fs ..
The uncertainties are added to the manuscript.

Remark: in a purely ballistic picture (as in the ballistic case in Ref 35), without introduction of
the orbital flip and the resulting deviation from linearity, the group velocity v_o is roughly given by
the inverse slope $\frac{\tau_D}{d_{Cu}}$ of Fig. 2e which gives a value of about $v_o = 0.1$ nm/fs (from the 4 last data
points) The orders of magnitude we obtain in both ways, i.e. the purely ballistic picture and the
ballistic picture with some small probability of orbital flip, are much smaller than that of the charge
velocity in Cu by about an order of magnitude. The reasons are discussed in the maintext.

Line 221-231

*The delay is expressed as a function of τ_{of} and $t_0 = \frac{d_{Cu}}{v_o}$, where v_o is the group velocity of the*
*orbital carriers, in the following expression:*

$$174 \quad \tau_D = \frac{\int_{t_0}^{\infty} dt \exp\left(-\frac{t}{\tau_{of}}\right)}{\int_{t_0}^{\infty} \frac{dt}{t} \exp\left(-\frac{t}{\tau_{of}}\right)} \quad (\text{Eqn. 2})$$

*As shown in Fig. 2e, the curves calculated from Eqn. 2 show a qualitative agreement with the*
*experimental variation for the deviation from linearity and for the concavity. The typical orbital group*
*velocity of carriers extracted from our fit, $v_o \approx 0.26 \pm 0.08$ nm/fs, appears to be an order of magnitude*
*below the charge carrier velocity in Cu at room temperature. This striking result is not really surprising*
*because charge currents and orbital currents cannot be carried by the same type of carriers and same*
*type of bands in the crystal. In the particular case of Cu, one knows that the charge current is*
*predominantly carried by s-band electrons (no orbital moment) whereas orbital currents are necessarily*
*carried by carriers of s-p hybridization with enhanced effective mass and possibly carriers at higher*
*energy induced by light.*

On line 197, it is written that “Another striking result...”. I do not understand what the authors had
in mind on the concave shape. This needs to be more specific. Thus, I do not understand how it is
not the expected behavior from the diffusive transport, which also needs to be mentioned.

**Response:**

This part of the text has been rewritten to avoid any confusion.

Moreover, the discussion of the expected behavior of the various transport phenomena is
explained in details in the Methods: Data Analyses section.

Line 216-224

*Our results in Fig. 2d rule out an interpretation in the diffusive regime. In the ballistic regime, we*
*extended the formalism in [35] by introducing a probability of orbital flip (or orbital decoherence) that*
*we characterize by the orbital flip time τ_{of} , (see Methods and Supplementary Information S1). The*
*probability of orbital flip, even small, introduces deviations from the linear variation of the delay as a*
*function of the thickness of Cu, see the concavity of the curves in Fig. 2e. The delay is expressed as a*
*function of τ_{of} and $t_0 = \frac{d_{Cu}}{v_o}$, where v_o is the group velocity of the orbital carriers, in the following*
*expression:*

193

$$\tau_D = \frac{\int_{t_0}^{\infty} dt \exp\left(-\frac{t}{\tau_{of}}\right)}{\int_{t_0}^{\infty} \frac{dt}{t} \exp\left(-\frac{t}{\tau_{of}}\right)} \quad (\text{Eqn. 2})$$

*As shown in Fig. 2e, the curves calculated from Eqn. 2 show a qualitative agreement with the*
*experimental variation for the deviation from linearity and for the concavity.*

*Line 327-345*

*Such delay time depends on the Cu thickness (d_{Cu}) and the group velocity of orbital carriers*

*(v_0) according to $\tau_D(\cos\theta) = \frac{1}{v_0 \cos\theta} d_{Cu}$ in the case of a ballistic transport (d_{Cu} smaller than the*

*mean free path, MFP λ). Concerning the transport properties, one can then make the classification*
*below:*

- *a ballistic transport in the limit of a short coherence time where the traveling distance is linearly*
*proportional to the delay time $\tau_D \sim d_{Cu}$. This case is described in large details in Ref. [35].*

- *a diffusive transport where the average escape distance from the origin increases as the square root*
*of the time according: $d_{Cu} \sim \sqrt{D\tau_D}$ (linked to the diffusion constant D). This leads to a convex*
*shape of the time delay $\tau_D \sim (d_{Cu})^2$ vs. the Cu thickness d_{Cu} .*

- *a ballistic transport accompanied by a long decoherence process where the time delay increases*
*with the distance to the origin as: $\tau_D \sim (d_{Cu})^\alpha$ with ($\alpha < 1$) associated to a concave shape of*
*the corresponding dependence of τ_D vs. d_{Cu} .*

*Note that, in the case of hot carrier transport, there should be an intermediate regime of transport,*
*the so-called superdiffusive regime, intermediate between purely ballistic and diffusive regime ($1 < \alpha <$*
*2) corresponding to excitation of secondary source of electrons. Electrons may transfer their energy to*
*other electrons occupying states below the Fermi energy.*

Finally, the authors adopt the method of analysis from Ref. 35, which needs to be mentioned. Also,
it needs to be specified that how much the analysis in this manuscript is different from or share some
common features with that of Ref. 35.

**Response:**

We specify in the manuscript and Supplementary Information that the express of τ_D is derived
by extending the calculation proposed by Seifert et al.

*Line 217-219*

*we extended the formalism in [35] by introducing a probability of orbital flip (or orbital*
*decoherence) that we characterize by the orbital flip time τ_{of} , (see Methods and Supplementary*
*Information S1).*

Overall, I think this section needs to be written in a clearer way.

**Response:**

We realized there existed several inconsistencies which make this section not easy to follow.

We made the following changes to improve the clarity:

1. We use d_{Cu} instead of t_{Cu} to stand for the thickness of the Cu layer, which follows the
notation in the Section of Data analyses of METHODS.

2. To avoid confusion, we take extra care to make sure that the variation of τ_D vs. d_{Cu} is
always presented using τ_D as a function of d_{Cu} , instead of d_{Cu} as a function of τ_D , in agreement
with the experimental data shown in Figure 2e.

5. It needs to be clarified whether the polarization of the pump laser influences the result. For
example, see T. J. Huisman et al. Nature Nanotechnology 11, 455 (2016).

**Response:**

*Following the method used by Huisman et al, we pump the samples with femtosecond laser of*
*σ^+ , σ^- , and linear helicities, and recorded the terahertz emission. Figures S3 a and b show time traces*
*of the x component (a: perpendicular to the magnetic field) and the y component (b: parallel to the*
*magnetic field) of the terahertz emission for σ^+ , σ^- , and linear helicities. We observe that both the x*
*component and the y component remain the same, regardless of the laser helicity.*

Figure S3 The x component (a: perpendicular to the magnetic field) and
the y component (b: parallel to the magnetic field) of the terahertz emission
induced by pump lasers of σ^+ , σ^- , and linear helicities.

Line 102-104

*The polarization of the pump laser does not influence the THz emission from Cu/Ni samples (Fig.*
*S3 in Supplementary Information)*

6. On the terminology, I think using “orbital current” might be a better choice instead of “orbit
current” because the community already uses the term “orbital current” in many papers.

**Response:**

We agree with the reviewer and adopt the term “orbital current” for homogeneity.

Reviewer #2 (Remarks to the Author):

Authors: Yong Xu et al.

Title: Orbitronics: Light-induced Orbit Currents in Terahertz Emission Experiments

In this work the authors study THz emission from spintronic heterostructures. By comparing CoFeB
based and Ni based stacks and in combination with various normal metals, the authors identify
unexpected signals which they attribute to orbital momentum currents and to orbital-to-charge
conversion. This work is extremely timely, the data is clean and nicely presented and the results will
be of high interest for the community. This type of work would fit perfectly the scope of Nature
Communications. However, there are a number of sections of the manuscript that I believe are not
entirely clear and should be addressed before consideration for publication:

AHE1. On line 167 it is said «AHE contribution (IS) negligibly dependent on (THE) Cu thickness
as supported by the negligible departure from the signal with CoFeB monolayer in Cu/CoFeB
samples with different Cu thicknesses in our thickness range (Supporting information and referred
to above). » I have a few comments concerning this sentence:

1a. First of all the wording of the sentence is not very clear. What does “negligible departure from
the signal” mean here? I’m assuming they mean the signal is constant in Cu/CoFeB samples with
different thicknesses, which they attribute to the AHE-THz emission, and they assume it should be
the same for Cu/Ni... is it?

**Response:**

Yes. In the preceding version, considering that the emission with Cu/CoFeB, ascribed to AHE,
was very weakly dependent on Cu thickness, we extended this result to Cu/Ni(10 nm). For the new
version, we fabricated a series of Cu/Ni(2 nm) samples. With only 2nm, the AHE contribution
becomes negligible, as we measure on Ni(2 nm) single layers. Such a drop of the AHE contribution
in very thin magnetic layers is theoretically expected and experimentally observed in single
magnetic layers as well as in bilayers with a nonmagnetic metal (Ref 39, Mottamchetty et al). With
Cu/Ni(2nm) series having negligible contributions from AHE, the dependence on Cu thickness is
used to characterize the orbital contribution.

Line 188-209

*In Fig. 2c, we present and discuss the variation of the emission peak amplitudes for both series,*
*MgO/Cu/Ni (2 nm) and MgO/Cu/Ni (10 nm). For MgO/Cu/Ni (2 nm), the amplitude is about zero for*
*$d_{Cu} = 0$ (negligible AHE contribution with Ni 2 nm), and jumps to about its maximum value as soon as*
*a thin layer of Cu (2 nm) is deposited to cover almost totally MgO and then remains at about the same*
*level. The decay above 4 nm is due to the change in laser absorption and thin film impedance. For*
*MgO/Cu/Ni (10 nm) on the same figure, the negative amplitude at $d_{Cu} = 0$ expresses the negative*
*contribution from AHE and the positive contribution introduced by Cu reverses the signal as soon as the*
*thinnest Cu layer (1nm) is deposited to fully cover MgO before a decrease above 2 nm. The difference*
*with respect to the results for Ni (2 nm) is that this decrease can be due to the variation of the film*
*impedance as well as the variation of the AHE contribution (although the results presented in*
*Supplementary Information S5 for MgO/Cu/CoFeB seem to indicate that the insertion of Cu does not*
*change significantly the AHE, at least with CoFeB). So, the main result is that the signal immediately*

introduced by Cu, with Ni 2 nm as well as for Ni 10 nm, cannot be due to a conversion by IOHE in the
bulk of Cu, what would lead to a progressive increase of the signal following the increase of active region
for IOHE at increasing thickness of Cu. In addition, in Fig. 2d, we do not see some increase in the width
of the waveform as would be expected with contributions coming from different delays at different depths
in Cu layer with IOHE. The thickness dependence in Fig. 2c is characteristic of THz emission by the
interfacial orbital to charge conversion of light-induced orbital current by IOREE at the MgO/Cu
interface, an interfacial conversion already observed, for example, by Kim et al. [41] in several types of
trilayers including Co/Cu/MgO with the orbit current converted from charge current at Cu/MgO
interface and detected by torque on Co.

1b. Where is the Cu/CoFeB data? It is not in the Supporting information as indicated in the text. Is
it just the Cu/CoFeB vs CoFeB monolayer data from Fig1? Two data points is not a lot to claim an
“independence” on thickness...

**Response:**

We had by error removed the data for Cu/CoFeB at different Cu thicknesses in the
Supplementary and we have reintroduced them in the Supplementary of the new version. We find
that the introduction of Cu negligibly changes the terahertz emission of the CoFeB single layer.
However, as written a few lines above in the response to 1a, with the introduction of a new series
with Cu/Ni(2nm), the data on Cu/CoFeB are no longer useful for the interpretation of Cu/Ni and
simply an element of information on the AHE contribution with bilayers FM/NM.

We add the following to the Supplementary Information. Line 109-110:

*Figure S5 shows the THz emission from MgO/Cu/CoFeB(10 nm). The terahertz emission from*
*Cu/CoFeB samples remains roughly constant for the Cu thickness less than 6 nm (Figure S5).*

1c. What is the physical understanding of a Cu thickness independent AHE-driven THz emission?
That is pretty surprising a priori, since adding metal to the stack should change drastically the THz
(absorption) emission. Do the authors expect the AHE to increase with thickness, and therefore to
balance out the change in impedance of the stack? If not, what is the intuitive way of understanding
such an independence? Would this be the same for Ni and CoFeB samples? Could it be that in the
Cu/CoFeB layers there is also orbital currents going on?

**Response:**

As discussed in the response to 1a, the introduction of samples with very thin Ni and negligible
AHE contribution exempts us from referring to some similarity between the AHE contributions in
CoFeB and Ni.

About Cu/CoFeB, we agree with the reviewer that Figure S5 indicates that the AHE should
increase with the Cu thickness and balance out the change in impedance of the stack. The increase
of AHE with the Cu thickness might be due to the spin-current proximity effect reported by Dang
et al. (Dang et al. Phys. Rev. B 102, 144405 (2020)).

Figure R 3 The absorption of laser energy in CoFeB is shown as a function of the Cu thickness for Cu/CoFeB samples.

The reviewer raised another question about the possible orbital effects in MgO/Cu/CoFeB. Based on our results in Fig 1 the extremely small differences between MgO/CoFeB and MgO/Cu/CoFeB exclude the existence of significant orbital effects, from IOHE as well as the orbital-to-charge conversion at the MgO/Cu interface. We expect the orbital effects from CoFeB to be much less pronounced than Ni.

We add the discussion about AHE in Cu/CoFeB in the Supplementary Information. Line 111-113:

Terahertz emission from MgO/Cu/CoFeB is mainly driven by AHE. This means that, at least for CoFeB, adding Cu layers to a magnetic layer in this thickness range, does not change significantly the AHE contribution to THz emission.

The assumptions made in the paper are pretty strong, and not discussed enough. In the present form I believe that the amplitude of the orbital effect cannot be properly separated from that from the AHE, and unless having more convincing evidence the thickness dependence of the orbital effect can only be speculatively discussed.

Response:

As written in the response to 1a, to avoid the problem of the separation between orbital and AHE, we have prepared a new series of Cu(2nm)/Ni sample in which the AHE contribution becomes negligible. The AHE-induced THz emission from the 2nm Ni single layer as theoretically expected with very thin magnetic layers as well as experimentally observed in single magnetic layers or in bilayers with a nonmagnetic metal (ref 39, Mottamchetty et al).

1d. The conclusion on line 169 is not entirely obvious in my opinion. I believe it is a far stronger argument the one used earlier (lines 121-131...) where the polarities expected from SHE and other effects are discussed, and used to suggest that an additional phenomena is required (such as orbital currents).

Response:

We agree with the review that the message delivered on line 169 is stronger than line 121-131.
In the previous version, we discussed on lines 121-131 the possible channels of FM/NM/oxide
heterostructures to create the charge current that leads to THz emission, including AHE, spin-to-
charge conversion (shown in Figure 2a), and orbital-to-charge conversion (shown in Figure 2a).
On line 169, we consider only the AHE, spin-to-charge conversion and the orbital-to-charge
conversion (shown in Figure 2b) for Ni/Cu/MgO.

(1) AHE contribution has been discussed previously (question 1c).

(2) Spin-to-charge conversion. Cu is known for a small spin Hall angle. Moreover, the THz emission
from CoFeB and CoFeB/Cu(6 nm) are very similar, suggesting adding a Cu layer only introduce a
weak signal.

(3) Orbital-to-charge conversion. The important point is that the signal with Cu/Ni starts as soon as
the thinnest Cu layer (1nm) is deposited to cover almost totally MgO and then remains almost at
about the same level before decreasing slowly above 3nm. A conversion of the orbit current by
IOHE in the bulk of the Cu layer would not start abruptly but with a progressive increase of the
signal following the increase of active region for IOHE at increasing thickness of Cu. In addition,
we do not see some increase in the width of the waveform as would be expected with contributions
coming from different depths differently delayed in time in the Cu layer. The thickness dependence
in Fig.2c is characteristic of THz emission by orbit to charge interfacial conversion by IOREE at
the MgO/Cu interface with the expected additional slow decrease with thickness due to increasing
light absorption.

In short, we estimate that the dominant contribution for THz emission from MgO/Cu/Ni
samples arises from the conversion between charge and orbit at the MgO/Cu interface due to IOREE
(the conversion channel shown in Figure 2b), as observed from OREE at other MgO/Cu or oxide/Cu
interfaces (Kim et al., arXiv.2307.09824, 2023).

The dominant contribution is discussed in Lines 171-187:

*For a quantitative analysis of the different contributions in Eqn.1, MgO/Cu/Ni is our simplest system*
*for the following reasons.*

*a) Due to the very small SOC of Cu, we can assume that the ISHE in Cu and ISREE at the MgO/Cu*
*interface have negligible contributions, as expected for ISHE from the negligible calculated Spin Hall*
*Conductivity in Cu [40] and, for ISREE, because its contribution should have appeared after inserting*
*Cu between MgO and CoFeB (Fig. S5).*

*b) The respective contributions from IOHE in Cu and IOHEE at the MgO/Cu interface can be*
*distinguished from the dependence on Cu thickness in a series of MgO/Cu/Ni samples, as we will see*
*below. In similar Co/Cu/oxide structures, including Co/Cu/MgO, the predominant orbital effects have*
*been clearly ascribed to the Cu/oxide interfaces [41].*

*c) Although the AHE contribution in Ni 10 nm single layers is small enough to be reversed by*
*inserting Cu (Fig.1b), its dependence on Cu thickness could introduce uncertainties in the discussion of*
*the dependence on Cu thickness in the MgO/Cu/Ni (10 nm) series. For this reason, we studied a similar*
*series with Ni (2 nm) for which we found a negligible small AHE-induced THz emission from a single*
*layer of Ni 2 nm (Section 4 of Supplementary Information). The drop of the AHE contribution at small*
*Ni thicknesses is consistent with theory and previous observations for the emission from AHE in ultra-*
*thin metallic films [38].*

2. On line 182 the authors say “with the expected additional slow decrease with thickness due to
increasing light absorption”. Have the authors estimated this decrease? They claim it is “slow”, but
how slow is it? Is it compatible with a ~50% loss of signal per 6nm of Cu as their data suggests?
From the literature, I would say it is much more...

**Response:**

According to the light absorption profile, we estimate ~10% loss of absorbed energy in CoFeB
(Figure R3).

The reviewer suggested the loss of signal should be more than 50% per 6 nm of Cu. In order
to examine the signal loss, we studied a new series of samples MgO/Cu(0-6 nm)/Ni(2 nm)/MgO
to demonstrate the loss of signal at 6 nm of Cu (Figure S4). The sample MgO/Ni (2 nm)/MgO shows
negligible THz emission, and can be taken as a good reference sample. The THz emission increases
to the maximum with Cu 2 nm, corresponding to a well-established MgO/Cu interface. The peak
amplitude of Cu 6 nm decreases to approximately 1/3 with respect to the peak amplitude of Cu 2
429 nm. We rewrite the sentence as follows. Line 192-193

*The decay above 4 nm is due to the change in laser absorption and thin film impedance.*

3. My final point is on the uncertainty in the reported delay times in Fig2.e. The measurements have
no X or Y error bars (neither in Fig2d nor Fig2e). The delay time seems to be “picked” by choosing
the data point at the highest amplitude and not by any fitting function. Also, the fact that a 35fs pulse
is being used to extract data points with a ~5fs time-resolution is not discussed. In fact, if I put an
error bar equal to half the FWHM (~17fs) onto the data points in Fig.2e I could fit almost any
function. The authors should add error bars if they want to discuss such measurements and should
discuss the technical details to achieve such a reduced uncertainty. A few points the authors might
want to clarify:

The data points in Fig2.d the result of single traces or are they the result of multiple scans, in which
case could they could show the statistics? Have the authors carried some noise characterization in
their setup (amplitude and timing noise)?

**Response:**

The data points in Fig2.d are the result of single traces. We performed multiple scans to address
the concerns of the reviewer. The Fig. R4 shows the THz emission signal for substrate//MgO/Cu/Ni
(10 nm) samples for different Cu thicknesses obtained from average 5 scans. Based on the multiple
scans, our waveforms have a noise of ~0.2uV at the peak signal, in agreement with the noise level
(~0.2 uV) of our setup.

From the multiple scans of a given sample, we estimate the error bar in the time domain to be
less than ~5 fs.

Figure R 4 Noise characterization of (a) the amplitude and (b) the time
 delay.

How many time constants did the authors wait before pulling the measurements from the lock-in,
 and what filters (or ENBW) were they using?

**Response:**

We use a lock-in SR830 to measure the signal which is modulated by a chopper of ~ 200 Hz.
 The lock-in SR830 extracts the signal at the modulation frequency, with a detection bandwidth
 inverse proportional to the time constants. Typically, the time constant is set to 300 ms. An interval
 of 1000 ms is adopted to stabilize the mechanical movement of the delay line.

Those experimental details are added to the Methods Section: Experiment Procedure (Line
 318-322).

Are the measurements independent of delay-line speed, or in other words did the authors carefully
 verified that the measurements at each position are independent of previous positions? One neat
 way of demonstrating that the delay line and lock-in timings are just right is to perform the
 experiments by randomizing the order of time-delays to measure (for ex: measuring first -1ps, then
 +0.5ps, 0ps, -0.3ps... and so on randomly, instead of going from -2ps to +0.5ps in incremental order).
 I believe that all of these points could potentially strongly affect the confidence in the observed
 delays. With proper error bars, a discussion becomes possible which would, indeed, be very
 interesting.

**Response:**

We thank the reviewer for the suggestion. We performed the experiments by randomizing the
 order of time steps and the results is shown in Fig. R5. We make sure the setup waits long enough
 before measuring the signal, so that the measurement is independent of the delay-line speed. The
 waveform obtained using the random time steps (open square) is almost the same as that obtained
 using incremental time steps (line). It can be understood because we use an incremental step size of
 0.005 mm for the delay line, which is one order of magnitude larger than the positioning resolution
 of the delay line (0.0003 mm).

Figure R 5 Comparison between the time traces of emitted terahertz waves obtained by randomizing time delays and incremental time delays.

Finally, we also show the experimental results obtained on non-orbital-Rashba systems CoFeB/Ta giving rise to bulk spin-ISHE and used as a 'reference system' does not show any time shift with the same experimental protocol on the same optical experimental bench (Fig. R6).

Figure R 6 Experimental results obtained on non orbital-Rashba systems

Those results are added to the Section 6 of Supplementary Information.

Reviewers' Comments:

Reviewer #1:

Remarks to the Author:

I thank the authors for kindly answering to all my questions and for considering my suggestions. I find the manuscript is also revised accordingly. I recommend the manuscript for publication.

Reviewer #2:

Remarks to the Author:

The manuscript is improved and I believe it should be published.

I appreciated the measurements taken to demonstrate the resolution of the setup. The authors seem to have extremely high SNR and very low timing jitter, and can reliably find the same results. I agree that the time resolution of their setup is probably much better than the pulse duration.

Nevertheless, I remain skeptical about some of the analysis and the strong conclusion on IOREE as the only possible mechanism to explain some of the signals. I would suggest the authors to remind the reader that this is a somewhat speculative discussion given the complexity of the problem, that numerous unknowns remain, and to let the question a bit more open.

However, there are some points I do not fully grasp:

-When I look at the delay of the Ni(10), it appears to me that the delay is longer than that for Cu/Ni(10) samples. How can one understand this?

-The authors studied the 2nm Ni film where they argue there is no AHE. In that case they can discuss amplitude variations in the signal. However, they do not plot the delays with those films in Fig.3d, only the 10nm which are mixed with the AHE. If the AHE has some distinct delay from the orbital part, say much slower, how would increasing the thickness of Cu thickness affect the observed delay overall? It seems very difficult to attribute the observed delay to a single effect (orbital) when we know that the signal is coming from two different sources, and the time delays of the first one is not considered...

-The authors argue in line 203 that "the do not see an increase of the width of the waveform", which confirms the I. However, if the speed is indeed 0.3 nm/fs, 6nm of Cu would only lead to ~23 fs of increase, which is impossible to tell on Fig.2d. If the authors want to make that claim, they should extract and plot FWHMs or FFTs as a function of thickness.

-In line 201 the authors say that if the effect was due to some Hall-like bulk effect (namely the IOHE) then the increase of thickness should lead to an increase in signal. However, the observation of the amplitude with thickness is pretty much the same that is observed in the case of ISHE, where the rapid drop in signal with metal thickness (impedance change) basically overtakes any possible growth in signal due to the ISHE. As an example, the authors can plot the amplitude vs thickness of the CoFeB/Ta samples that they added to their rebuttal, and they will see that the amplitude drops with thickness, even though it is due to ISHE.

On line 177 the authors write IOHEE, but I think they mean IOREE. The acronyms get confusing at times since there are so many. Maybe the authors can think of a simplified way of making reference to the various effects, or directly write Orbital Rashba and Orbital Hall in plain text.

REVIEWER COMMENTS

Reviewer #1 (Remarks to the Author):

I thank the authors for kindly answering to all my questions and for considering my suggestions. I find the manuscript is also revised accordingly. I recommend the manuscript for publication.

Response:

We thank the reviewer for recommending the manuscript for publication.

Reviewer #2 (Remarks to the Author):

The manuscript is improved and I believe it should be published.

I appreciated the measurements taken to demonstrate the resolution of the setup. The authors seem to have extremely high SNR and very low timing jitter, and can reliably find the same results. I agree that the time resolution of their setup is probably much better than the pulse duration.

Nevertheless, I remain skeptical about some of the analysis and the strong conclusion on IOREE as the only possible mechanism to explain some of the signals. I would suggest the authors to remind the reader that this is a somewhat speculative discussion given the complexity of the problem, that numerous unknowns remain, and to let the question a bit more open.

Response:

We thank the reviewer for the suggestion. We added a few words to address the concern of the reviewer in text of the new version. Line 257-260:

In light of the complexity arising from different contributions, we remind the reader that our theoretical discussion considered only the main contribution, the orbital-to-charge conversion at the Cu/MgO interface.

However, there are some points I do not fully grasp:

1. When I look at the delay of the Ni(10), it appears to me that the delay is longer than that for Cu/Ni(10) samples. How can one understand this?
2. The authors studied the 2nm Ni film where they argue there is no AHE. In that case they can discuss amplitude variations in the signal. However, they do not plot the delays with those films in Fig.3d, only the 10nm which are mixed with the AHE. If the AHE has some distinct delay from the orbital part, say much slower, how would increasing the thickness of Cu thickness affect the observed delay overall? It seems very difficult to attribute the observed delay to a single effect (orbital) when we know that the signal is coming from two different sources, and the time delays of the first one is not considered...

Response to 1: The comparison of the delays with Ni (10 nm) and Cu/Ni (10 nm)

The THz emission by a single ferromagnetic layer (F) via AHE is due to a completely different mechanism compared to those involving spin to charge conversion in usual spintronic emitters or orbital current to charge conversion in the MgO/Cu/Ni structures. The AHE emission by a single F layer, say Ni, is related to a “backflow” charge current resulting from the different reflections of hot electrons on opposite interfaces (V. Mottamchetty, et al. Scientific Reports 13, 5988, 2023). As the emissions of Ni (10 nm) comes from AHE (that is from an internal backflow of charge current in F (Ni) then deflected by AHE to generate a transverse charge current), while that of Cu/Ni (10 nm) comes in a majority way (larger by a factor of two with opposite sign) from orbital currents in Cu converted into charge current at the Cu/MgO interface, there is no reason for comparing the delays in systems presenting physically different emission mechanisms.

Response to 2 about: They do not plot the delays for Ni 2 nm.

We agree with the reviewer that a study of the delay in the absence of AHE contribution (Cu/Ni (2 nm)) would be presently the best way. However, with only 2 nm for Ni in our Cu/Ni (2 nm), the signal becomes very small (Fig. S4 in Supplementary Information). It is thus difficult to extract a meaningful and reliable value of the delay due to the weak signal ~ 1.5 uV (THz waveforms have a noise of ~ 0.2 uV at the peak signal). We have nevertheless extracted the delay time and the result, with large error bars, is shown in the figure below, seeming to increase vs. Cu thickness in rough qualitative agreement with the ballistic behavior with Cu/Ni (10 nm), at least ruling out the quadratic increase of the diffusive regime.

Figure R 1 (a) THz emission signal for MgO/Cu/Ni (2 nm) samples for different Cu thicknesses. The maxima of the signals are highlighted by a circular marker. (b) Corresponding time delay versus Cu thickness d_{Cu} .

We added a few words in text of the new version. Line 220-221:

As the small signals obtained with MgO/Cu/Ni (2nm) are too small for a precise analysis, we consider only the delays for Ni (10 nm).

Response to 2 about: “the delay is coming from signals coming from both conversion of orbital currents outside Ni and AHE in Ni.”

For this reason of higher reliability, we did not use the delay data for Ni 2 nm (negligible AHE but too large noise on the delays) and we considered the delay data for Cu/Ni (10 nm). We agree that the emission not only comes from conversion of orbital current (about 2/3) but also from AHE, with AHE contribution smaller by a factor of two and opposite sign. We agree that we made the approximation that such AHE internal effects in Ni could not affect significantly the dependence of the delay on the Cu thickness, as also supposed in the previous work of T. Seifert et al. in the case of Ni/W system (Ref.[43] of our manuscript).

We added the following sentence in Line 231-233 of the new version:

We made the approximation that the AHE contribution in Ni 10 nm could not affect significantly the dependence of the delay on the Cu thickness.

3. The authors argue in line 203 that “they do not see an increase of the width of the waveform”, which confirms the I. However, if the speed is indeed 0.3 nm/fs, 6nm of Cu would only lead to ~ 23 fs of increase, which is impossible to tell on Fig.2d. If the authors want to make that claim, they should extract and plot FWHMs or FFTs as a function of thickness.

Response:

We thank the reviewer for the suggestion. We extracted FWHM and performed FFT on all the data in Fig. 2d. The FWHMs and FFT spectra are the same within the error range, showing that there is no increase in the width of the waveform. We also added these data to the Supplementary Information (Figure S6).

Figure R 2 (a) FWHMs and (b) Fourier spectra of the THz waveforms from Cu/Ni (10 nm) heterostructures with varying Cu thickness.

4. In line 201 the authors say that if the effect was due to some Hall-like bulk effect (namely the IOHE) then the increase of thickness should lead to an increase in signal. However, the observation of the amplitude with thickness is pretty much the same that is observed in the case of ISHE, where the rapid drop in signal with metal thickness (impedance change) basically overtakes any possible growth in signal due to the ISHE. As an example, the authors can plot the amplitude vs thickness of the CoFeB/Ta samples that they added to their rebuttal, and they will see that the amplitude drops with thickness, even though it is due to ISHE.

Response:

As we write in the new version, there are now stronger arguments to ascribe the emission in MgO/Cu/Ni samples to a predominant IOREE effects at MgO/Cu interfaces, much larger than the bulk IOHE. These arguments come from theoretical predictions of very small OHE in Cu [48], experimental evidence of predominance of IOREE at Cu/MgO interface over IOHE in Cu of thickness in the nm range [49], and the Cu thickness dependence (Fig. 2c).

Line 179-212:

For a quantitative analysis of the different contributions in Eqn.1, MgO/Cu/Ni is our simplest system for the following reasons.

a) Due to the very small SOC of Cu, we can assume that the inverse spin Hall effect in Cu and inverse spin Rashba-Edelstein effect at the MgO/Cu interface have negligible contributions, as expected for inverse spin Hall effect from the negligible calculated Spin Hall Conductivity in Cu [48] and, for inverse spin Rashba-Edelstein effect, because its contribution should have appeared after inserting Cu between MgO and CoFeB (Fig. S5).

b) From ab-initio calculations [48] the OHE of Cu is one of the lowest OHE in metals, whereas large OREE have been observed for the Cu/MgO interface [49] in Co/Cu/MgO trilayers which show the predominant effects of conversion from charge current to orbital current.

c) In addition, the predominant contribution from IOREE at the MgO/Cu interface over IOHE is consistent with the dependence of the emission amplitude on Cu thickness in MgO/Cu/Ni samples (Fig. 2c). For IOREE, one expects the abrupt increase as soon as MgO is covered by Cu, in agreements with Fig. 2c. In contrast, for bulk IOHE, one would expect a progressive increase following the increase of the active region. To avoid the interplay between the contributions from conversion of orbit current (2/3) and AHE (1/3 and opposite signs) in MgO/Cu/Ni (10 nm), we studied a similar series with Ni (2 nm) for which we found a negligibly small AHE-induced THz emission from a single layer of Ni 2 nm, as expected and observed in ultra-thin metallic films [46] (Section 4 of Supplementary Information). For MgO/Cu/Ni (2 nm) in Fig.2c, the amplitude is about zero for $d_{Cu} = 0$ (negligible AHE contribution with Ni 2 nm), and jumps to about its maximum value as soon as a thin layer of Cu (2 nm) is deposited to cover almost totally MgO and then remains at about the same level. The progressive decay above 4 nm is due to the change in laser absorption and thin film impedance. For MgO/Cu/Ni (10 nm) on the same figure, the negative amplitude at $d_{Cu} = 0$ expresses the negative contribution from AHE and the positive contribution introduced by Cu reverses the signal as soon as the thinnest Cu layer (1nm) is deposited to fully cover MgO before a progressive decrease above 2 nm. The difference with respect to the results for Ni (2 nm) is that this decrease can be due to the variation of the film impedance (light absorption) as well as the variation of the AHE contribution (although the results presented in Supplementary Information S5 for MgO/Cu/CoFeB seem to indicate that the insertion of Cu does not change significantly the AHE, at least with CoFeB).

So, the main result is that the signal abruptly reached at its maximum value by covering MgO with Cu, with Ni 2 nm as well as with Ni 10 nm, is not consistent with a progressive introduction by IOHE in the bulk of Cu and in favor of interfacial IOREE, as expected by theory and observed by Kim et al. [49] in several types of trilayers including Co/Cu.

5. On line 177 the authors write IOHEE, but I think they mean IOREE. The acronyms get confusing at times since there are so many. Maybe the authors can think of a simplified way of making reference to the various effects, or directly write Orbital Rashba and Orbital Hall in plain text.

Response:

We thank the reviewer to point out this mistake. We corrected it in the new version. To minimize the confusion caused by acronyms, we keep only IOREE, IOHE and AHE. The inverse spin Hall effect and inverse spin Rashba-Edelstein effect is written in plain text (except in Figures).

Reviewers' Comments:

Reviewer #2:

Remarks to the Author:

I thank the authors for the effort of providing new data and clarifying some of my questions.

I believe the manuscript is improved, and will undoubtedly trigger many interesting questions in the community.